# Assembly and lipid-gating of LRRC8A:D volume-regulated anion channels

Antony Lurie[1,2], Christina A. Stephens [3], David M. Kern[1,2,5], Katharine M. Henn [4], Naomi R. Latorraca [3] & Stephen G. Brohawn [1,2,4] ✉

Volume-regulated anion channels (VRACs) are ubiquitously expressed vertebrate ion channels that open in response to hypotonic swelling. VRACs assemble as heteromers of LRRC8A and LRRC8B-E subunits, with different subunit combinations resulting in channels with different properties. Recent studies have described the structures of LRRC8A:C VRACs, but how other VRACs assemble, and which structural features are conserved or variant across channel assemblies remains unknown. Herein, we used cryo-EM to determine structures of a LRRC8A:D VRAC with a 4:2 subunit stoichiometry, which we captured in two conformations. The presence of LRRC8D subunits widens and increases hydrophobicity of the selectivity filter, which may contribute to the unique substrate selectivity of LRRC8D-containing VRACs. The structures reveal lipids bound inside the channel pore, similar to those observed in LRRC8A:C VRACs. We observe that LRRC8D subunit incorporation disrupts packing of the cytoplasmic LRR domains, increasing channel dynamics and opening lateral intersubunit gaps, which we speculate are necessary for pore lipid evacuation and channel activation. Molecular dynamics simulations show that lipids can reside stably within the pore to close the channel. Using electrophysiological experiments, we confirmed that pore lipids block conduction in the closed state, demonstrating that lipid-gating is a general property of VRACs.

Volume-regulated anion channels (VRACs) are a family of ubiquitously expressed osmosensitive vertebrate ion channels that mediate cellular volume regulation and paracrine signaling[1,2]. VRACs are hexameric large pore channels that assemble as heteromers of LRRC8A and LRRC8B-E subunits[3–5]. With up to 1934 unique channel assemblies possible, VRAC assembly diversity is thought to account for the observed variability in channel properties, including single channel conductance, open probability, rectification, voltage-dependent inactivation, and substrate selectivity[3,5–11]. How differences in channel composition are structurally manifested to modulate channel properties remains largely unknown.

Recently, studies from our group[12] and Rutz et al.[13] described the first high-resolution structures of heteromeric LRRC8A:C VRACs. Both studies identified a similar hexameric channel architecture, but with different LRRC8A:C stoichiometries of 5:1 or 4:2, respectively. We additionally observed lipids bound inside the pore, which occlude the channel in the closed state. How other heteromeric LRRC8 channels assemble and whether they are similarly lipid-gated is unknown. Compared to other characterized VRAC assemblies (i.e., LRRC8A:C/E), LRRC8A:D VRACs have been reported to maintain smaller Cl⁻ conductances, greater open probabilities, and increased permeation of larger and non-anionic substrates, including taurine, γ-aminobutyric

[1]Department of Molecular and Cell Biology, University of California, Berkeley, CA, USA. [2]California Institute for Quantitative Biology (QB3), University of California, Berkeley, CA, USA. [3]Department of Biochemistry and Molecular Biophysics, Columbia University Irving Medical Center, New York, NY, USA. [4]Department of Neuroscience, University of California, Berkeley, CA, USA. [5]Present address: ModeX Therapeutics, Weston, MA, USA. ✉ e-mail: brohawn@berkeley.edu

acid (GABA), *myo*-inositol, cisplatin, and blasticidin, although the magnitude of these differences varies between reports[3,5–8]. By determining cryo-EM structures of a LRRC8A:D VRAC, we aimed to identify which features are shared with LRRC8A:C VRACs, while also establishing a structural basis for their unique channel properties.

## Results and discussion

To study LRRC8A:D VRACs by cryo-EM, we leveraged our previously described approach for characterizing fiducially-tagged LRRC8A:C VRACs (Supplementary Fig. 1)[12]. Mouse LRRC8A and LRRC8D subunits were over-expressed in insect cells (which lack endogenous LRRC8s) using a single engineered baculovirus. Each subunit was C-terminally tagged with a fluorescent protein fused through a protease-cleavable linker, with LRRC8A and LRRC8D subunits bearing different fluorescent proteins and protease cleavage tags to enable subunit-specific pulldown. To eliminate channel pseudosymmetry for cryo-EM studies, a BRIL domain[14,15] was inserted into the extracellular loop of LRRC8A. Fiducial mass was further increased by adding an α-BRIL antibody fragment (Fab)[16] and an α-Fab nanobody (Nb)[17] to the purified sample. Detergent solubilization, followed by two sequential rounds of affinity chromatography, allowed us to specifically isolate LRRC8A:D VRACs for cryo-EM single particle analysis.

We initially resolved a consensus map at an overall resolution of 3.1 Å (Supplementary Fig. 2), revealing a hexameric LRRC8 channel. However, poor map quality in three of the six subunits precluded confident assignment of each subunit's identity. Using 3D variability analysis in cryoSPARC[18], we resolved two conformations with

improved local map qualities to overall resolutions of 3.3–3.4 Å. These reconstructions allowed us to assign identities to all subunits and to construct atomic models incorporating the transmembrane domains, extracellular regions, and linker regions of all subunits (Fig. 1a, b and Supplementary Fig. 2 and 3). No N-termini (residues 1–14) were resolved inside the channel pore, presumably due to conformational heterogeneity. On the basis of fiducial and subunit-specific side chain densities (Supplementary Fig. 4), we inferred that both conformations correspond to the same channel assembly with a 4:2 LRRC8A:D stoichiometry and adjacent LRRC8D subunits. Our interpretation of these data is that the 4:2 LRRC8A:D assembly is the most prevalent in our purified sample. It is possible that additional assemblies are present, but are too structurally heterogeneous or rare to be resolved. Unless otherwise noted, we focus our analyses on the better resolved conformation 1.

Comparing LRRC8A:D to previously described structures of LRRC8A:C[12], LRRC8A homomers[19], and LRRC8D homomers[20] shows that LRRC8A:D follows a similar overall architecture, with a nearly invariant extracellular region (pairwise backbone r.m.s.d. <1 Å). Within this rigid extracellular scaffold, differences in selectivity filter residues may help explain differences in substrate selectivity between channel assemblies. The selectivity filter is formed by a single residue from each subunit pointing towards the conduction axis: R103 for LRRC8A, L105 for LRRC8C, and F144 for LRRC8D. Introduction of F144 residues from LRRC8D subunits increases hydrophobicity along one face of the filter compared to LRRC8A homomeric channels (Fig. 1c). The F144 side chain also permits more dilated filters since it lays flatter against

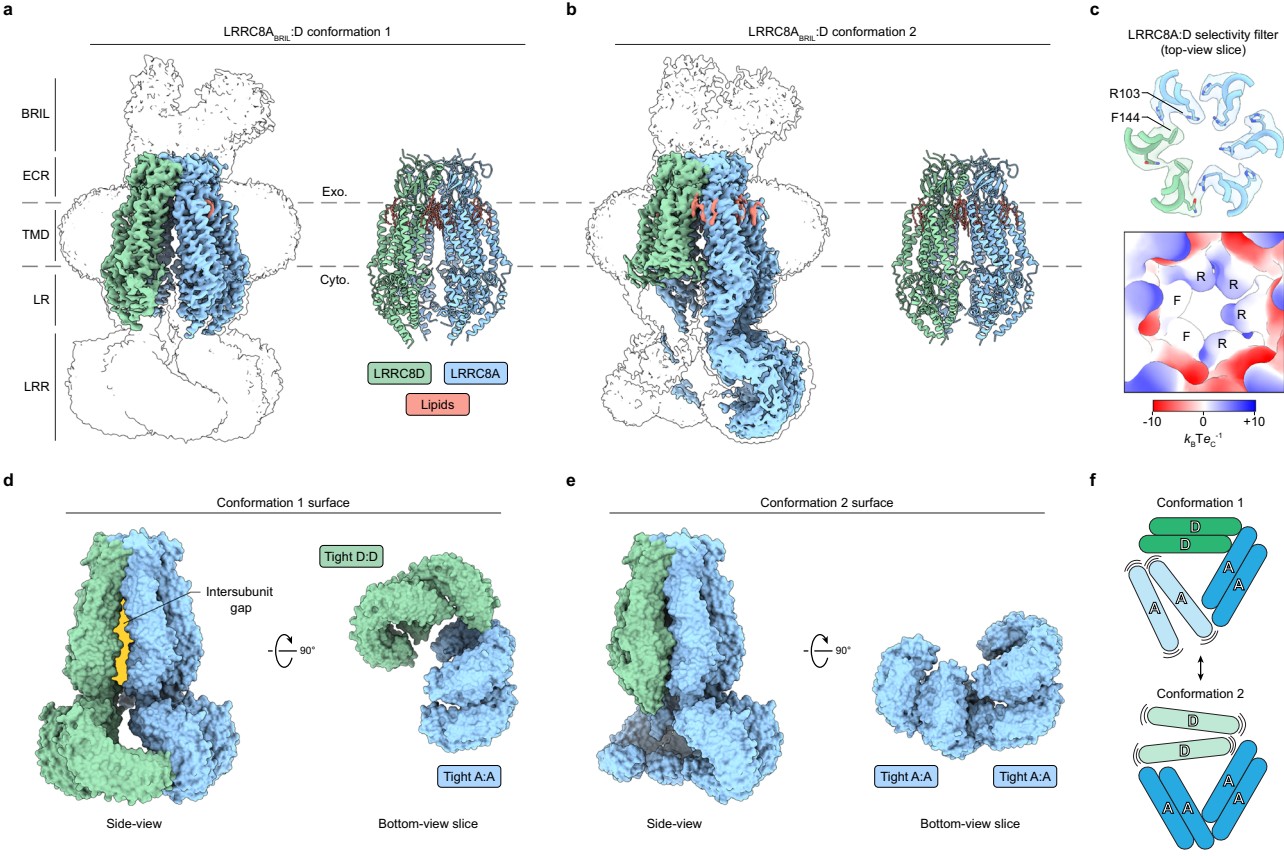

**Fig. 1 | Architecture of a LRRC8A_BRIL:D VRAC.** Side-views of cryo-EM density maps (*left*) and their associated models (*right*) for LRRC8A:D conformation 1 (**a**) and 2 (**b**). **c** Focused top-view of the LRRC8A:D selectivity filter model overlaid with its carved density (*top*) and the model surface colored by electrostatic potential (*bottom*). Side-view (*left*) and bottom-view slice (*right*) of the LRRC8A:D surface with docked leucine-rich repeat domains (LRRs) for LRRC8A:D conformation 1 (**d**) and 2 (**e**). The intersubunit gap induced by ordering of the LRRC8D LRR pair in conformation 1 (**d**, *left*) is highlighted in yellow. **f** Cartoon schematic of the LRRC8A:D LRR conformations. LRRC8A subunits, blue; LRRC8D subunits, green; lipids, salmon. Cyto cytoplasmic, ECR extracellular region, Exo exoplasmic, LR linker region, TMD transmembrane domain.

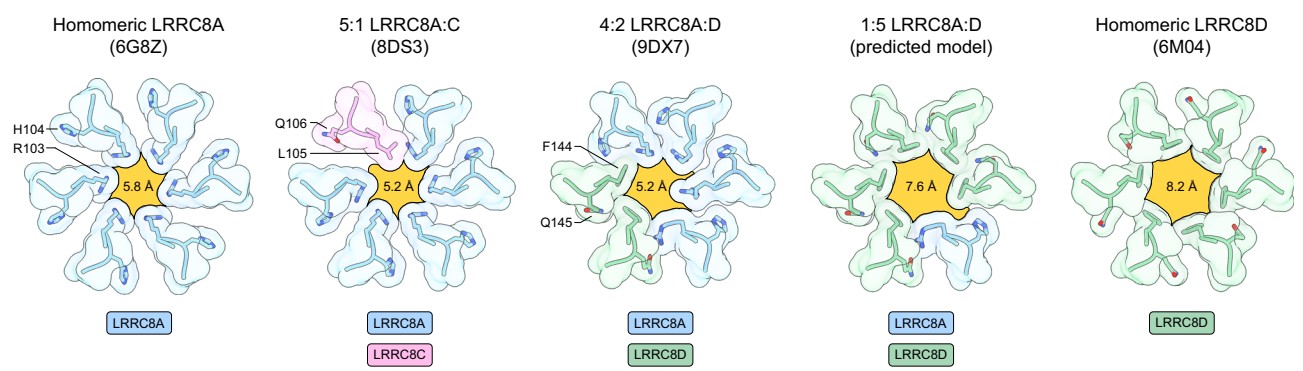

**Fig. 2 | Comparison of VRAC selectivity filters.** Focused top-views of selected selectivity filter models. The pore is outlined and highlighted in yellow and overlaid with the calculated minimum pore diameter. LRRC8A subunits, blue; LRRC8C subunits, pink; LRRC8D subunits, green.

the outer vestibule wall compared to L105 in LRRC8C or R103 in LRRC8A (Fig. 2). The structures of LRRC8A:D presented here are similarly constricted at the selectivity filter as LRRC8A:C (to ~5.2 Å in diameter for LRRC8A:D and LRRC8A:C), but this is due to an extended rotamer adopted by one LRRC8A R103 residue. We predicted that assemblies with higher LRRC8D to LRRC8A subunit ratios could house expanded selectivity filters. Indeed, a model of a 1:5 LRRC8A:D selectivity filter shows a predicted minimum pore diameter of ~7.6 Å, only slightly smaller than that observed in a structure of a non-physiological LRRC8D homomer (~8.2 Å). The expanded size and increased hydrophobicity of LRRC8D-containing channel filters are expected to promote, at least in part, their greater permeability to non-anionic or larger substrates, which is consistent with previous electrophysiology experiments that have established the importance of this site for channel selectivity[19–22]. Other regions along the conduction path may also help determine substrate selectivity, including the N-termini. While N-termini were not resolved in our LRRC8A:D structures, structures of LRRC8A[21] and LRRC8D homomers[20] have captured N-termini in pore-lining conformations, and prior electrophysiology experiments have evidenced that N-terminal residues influence substrate selectivity[21–23]. In particular, N-terminal residues 5 (T5 in LRRC8A-C, A5 in LRRC8D-E) and 8 (R8 in LRRC8A/C, K8 in LRRC8B/E, A8 in LRRC8D) may contribute to establishing the unique substrate selectivity of LRRC8D-containing VRACs.

The major structural differences between LRRC8A:D conformations and other VRAC structures arise from the cytoplasmic leucine-rich repeat domains (LRRs). Although poor local resolutions precluded their atomic modeling, the positions of four LRRs were sufficiently resolved to permit docking of LRR models for visualization purposes (Fig. 1d, e). Shared across both conformations, one pair of LRRC8A LRRs adopts an ordered "tight" interaction identical to those observed in structures of LRRC8A homomers[19,22,24] and LRRC8A:C heteromers[12,13]. In conformation 1, two LRRC8D subunits form a second tight LRR pair, similar to those observed in a homomeric LRRC8D structure[20], while the two remaining LRRC8A LRRs are disordered (Fig. 1d). The opposite is true in conformation 2, where the LRRC8D LRRs are disordered, but the two remaining LRRC8A subunits order to form a second tight LRR pair (Fig. 1e). These LRR dynamics can be conceptualized as a bistable equilibrium, with stabilization of a LRRC8A LRR pair in competition with, and alternating, with stabilization of the LRRC8D LRR pair (Fig. 1f). As in LRRC8A:C[12,13], incorporation of non-LRRC8A (i.e. LRRC8D) subunits prevents the LRRs from adopting the stable pseudo-symmetric trimer of dimers arrangement seen in structures of LRRC8A homomers[19,22,24]. Here, only four LRRs can adopt a stable position at a time.

Destabilization of the LRR arrangement by LRRC8D subunits propagates up the channel to alter the positions of linker regions and transmembrane domains. These changes are stark in conformation 1,

where ordering of the LRRC8D LRR pair causes one LRRC8D subunit to peel away from the conduction axis (Fig. 1d). This creates a large lateral intersubunit gap of ~6 Å or wider at the inner leaflet of the membrane; wide enough to permit passage of lipids into and out of the channel pore. Notably, our recent structures of a LRRC8A:C VRAC similarly show that incorporation of a LRRC8C subunit increases LRR dynamics and opens a lateral intersubunit gap capable of passing lipids, suggesting that these properties may be necessary for channel function[12]. One possibility is that lateral gaps stabilized by heterotypic LRR interactions may allow for the removal of pore-occluding lipids (discussed below), which would be impeded in a rigid channel closed off from the membrane. This may help explain why incorporation of non-LRRC8A subunits is necessary for full channel activation[3,5] as LRRC8A homomeric channels show reduced LRR dynamics, maintain small intersubunit gaps, and generate only small currents in cells.

We observe density inside the LRRC8A:D channel pore consistent with lipids that occlude the conduction axis, reminiscent of those found in LRRC8A:C and chimeric LRRC8A-C structures (Fig. 3 and Supplementary Fig. 4d)[12,25]. We modeled this density as three 1,2-dioleoyl-*sn*-glycero-3-phosphoethanolamine (DOPE) molecules, which bind in a ring-like conformation. The three lipids bind in an upright conformation and are pinched together by the C-terminal ends of each subunit's transmembrane helix 1 (TM1). The phospholipid headgroups are exposed to an extracellular-facing vestibule between the C-terminus of TM1 and the selectivity filter, while the hydrocarbon tails point intracellularly into the channel pore, often packing in hydrophobic grooves formed between subunits. The density for pore lipids is varied and weaker than adjacent regions for the protein, indicating that pore lipids are likely flexible and conformationally variable within the pore. Our structures suggest that most phospholipids could be accommodated, perhaps with the exception of those with larger headgroups, such as phosphatidylinositols and glycolipids. The presence of pore lipids, as modeled, constricts the pore to a radius of 1.5–1.6 Å and is expected to result in channel closure (Fig. 3c).

To investigate the stability of lipids inside the channel pore, we carried out all-atom molecular dynamics simulations initiated from the LRRC8A:D conformation 1 structure (Fig. 4 and Supplementary Figs. 5 and 6). This fully atomistic representation is required to accurately model non-covalent interactions between the bound lipids, the protein, and water in the pore. We elected not to include the LRRs in our simulations, as they were not fully resolved in the structure (Supplementary Fig. 5a). We retained the three modeled pore lipids as DOPE and embedded the protein in a 9:1 ratio of 1-palmitoyl-2-oleoylphosphatidylcholine (POPC) and cholesterol (see Methods). We initiated six independent simulations, resulting in 6 μs of unbiased simulations. In one simulation, the three pore lipids remained close to their initial positions over the course of the 1-μs simulation, constricting the pore to a radius smaller than that of a bare Cl⁻ ion (<1.8 Å),

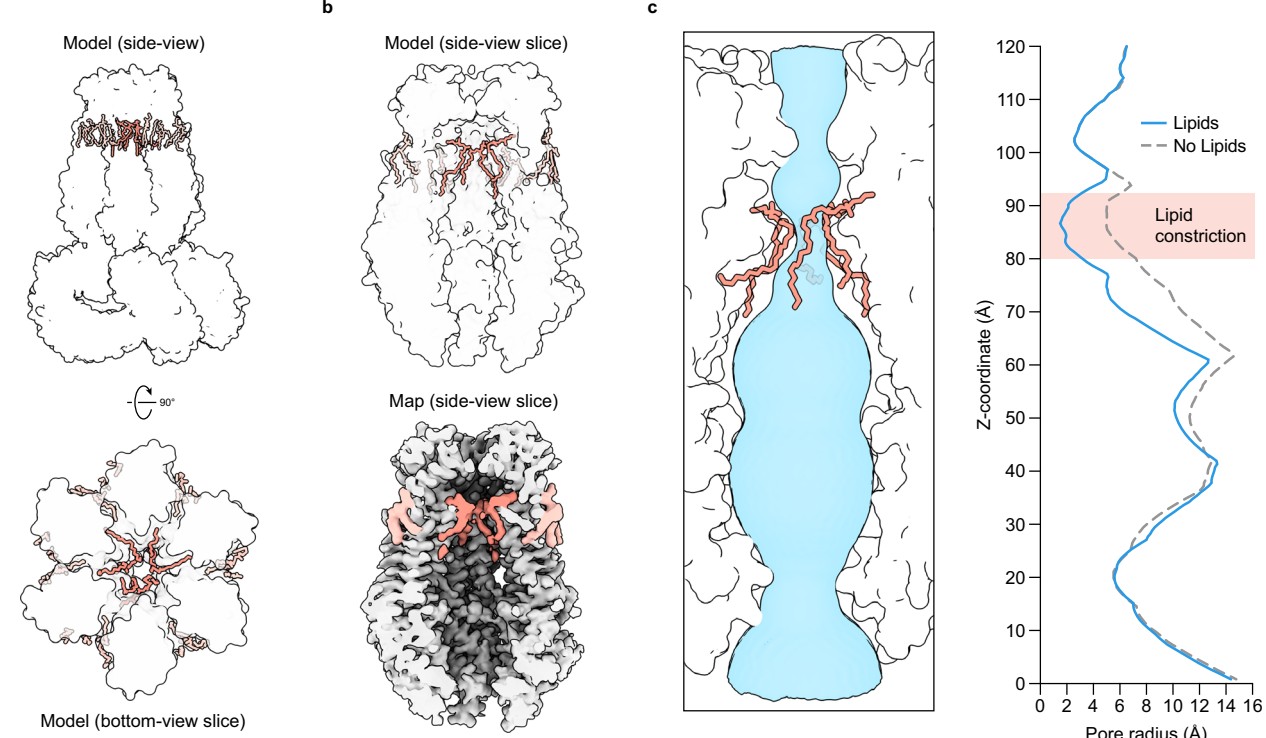

**Fig. 3 | LRRC8A:D VRACs are closed by pore lipids. a** Side-view (*top*) and bottom-view slice (*bottom*) of the LRRC8A:D model with pore (salmon, darker) and annular (salmon, lighter) lipids highlighted. **b** Side-view slices of the LRRC8A:D model (*top*) and carved map density (*bottom*) with pore and annular lipids highlighted. **c** *Left*: side-view slice of the LRRC8A:D model pore with lipids and the pore surface (blue) highlighted. *Right:* Calculated pore radii in the presence (blue, solid line) and absence (gray, dashed line) of pore lipids.

consistent with channel closure (Fig. 4, Supplementary Fig. 5c and Supplementary Movie 1). In the five remaining simulations, lipids dissociated from their starting positions in tens to hundreds of nanoseconds, accompanied by an increase in pore radius and pore water occupancy (Fig. 4, Supplementary Fig. 5c, d and Supplementary Movie 2). Pore radius measurements indicate that retaining at least two lipids close to their initial positions is sufficient for channel closure. To test if the initial configuration of bulk lipids may have impacted stability of the protein or the pore lipids, we ran six additional independent replicas with the protein embedded in a pure POPC bilayer for 1 μs each. Pore lipid stability was similar to that seen in the mixed bilayer, and we also observed one simulation where pore lipids did not dissociate from their initial binding positions (Supplementary Fig. 5b, c).

In all simulations, we observed that the pore lipid tails either packed against the pore-lining helices or hydrophobic intersubunit gaps, while the pore lipid headgroups remained near the center of the pore where they were well hydrated (Fig. 4b and Supplementary Movies 1 and 2). When pore lipids dissociated from their starting positions, the lipid tails retained these favorable hydrophobic interactions, while lipid head groups retreated intracellularly within the pore, sometimes even flipping their orientation (Supplementary Movie 2). Bulk lipid tails could also access the intersubunit gaps, suggesting a potential pathway for lipids to evacuate from the channel pore, although we did not observe lipids exchanging in and out of the pore on the timescales of our simulations.

We asked whether conformational changes in the protein relative to its starting structure could be leading to lipid dissociation events in the simulations. To test this, we initiated a third set of simulations with gentle harmonic restraints applied to the protein backbone in order to restrict deviation from the initial cryo-EM structure (Fig. 4, Supplementary Fig. 5c, d and Supplementary Movie 3). All three pore lipids remained close to their starting positions within the pore over the

course of these six 1-μs simulations, in good agreement with the starting cryo-EM-derived structure.

We found that in simulations without backbone restraints, LRRC8D subunits consistently rotated such that the large intersubunit gap between LRRC8A and LRRC8D subunits compacted, while the interfaces between LRRC8A subunits remained similar to the starting configuration (Supplementary Fig. 6a, b). These conformational changes did not obviously alter the radius of the channel pore (Supplementary Fig. 5c). Thus, lipids may disengage from their initial positions in the majority of our unrestrained simulations, not because of large-scale protein movements, but because of stochastic thermal motions in the protein altering the interactions between lipids, water, and protein within the pore. This hypothesis is consistent with the observation that mutations in the LRRs and linker regions, which increase channel dynamics, including recently described patient mutations, can result in constitutively active channels[11,12,26].

To experimentally test whether the observed pore lipids contribute to channel gating in LRRC8A:D VRACs, we used whole-cell patch clamp electrophysiology to characterize hypotonicity-induced currents from wild-type and mutant LRRC8A:D channels expressed in *LRRC8A-E*[-/-] HeLa cells (Fig. 5 and Supplementary Fig. 7). We compared wild-type channels to those with a T48D mutation in LRRC8A and/or LRRC8D subunits, which is predicted to electrostatically impede lipid occupancy in the channel pore (Fig. 5a). We previously characterized a T48D mutation in LRRC8A:C VRACs and found that it results in constitutive channel activity[12]. Cryo-EM structures of a LRRC8A(T48D):C channel showed that the mutation resulted in a loss of pore lipids without altering the channel's overall structure, stoichiometry, or conformational states. This is consistent with the T48D mutation increasing channel activity directly by disrupting pore lipid occupancy.

In cells transfected with wild-type LRRC8A and LRRC8D subunits, we observed robust channel activation (mean fold activation ± s.e.m. =

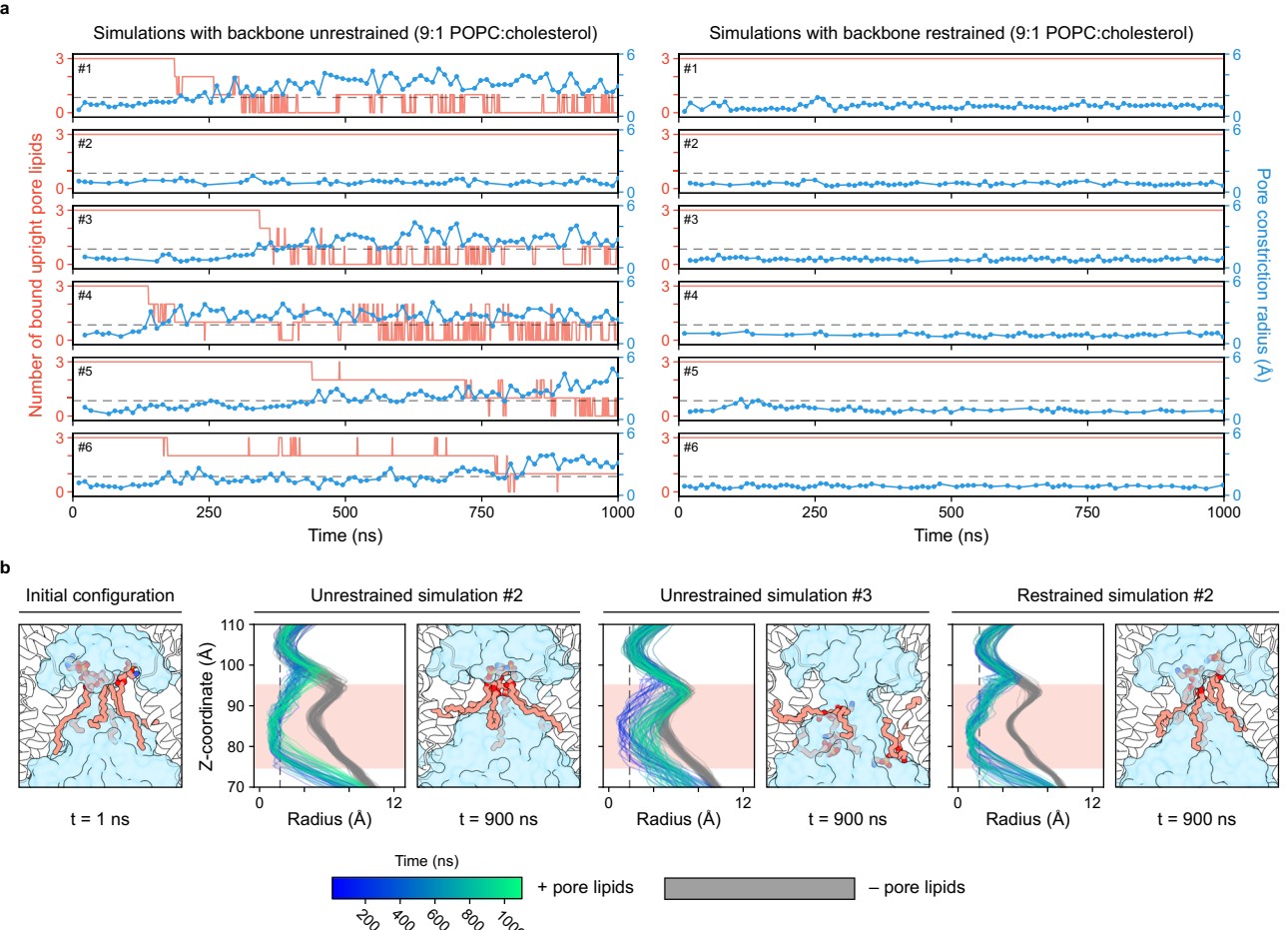

**Fig. 4 | Pore lipids can reside stably within the pore. a** Time traces from backbone unrestrained (*left*) and backbone restrained (*right*) simulations in 9:1 POPC:cholesterol membranes, indicating the number of lipids that remain bound upright within the pore (salmon; simulations sampled every 1 ns) and the measured radius of the pore (Å) at the constriction site (blue; simulations sampled every 10 ns, with missing time points reflecting structural snapshots that lacked plausible pores, as described in the Methods). The radius for a bare Cl⁻ ion (1.8 Å) is indicated with a gray dashed line. Individual simulation replicates (#1–6) are plotted successively from top to bottom. **b** Side-views of the pore immediately after the end of equilibration (*left*, 1 ns) and near the end of the total simulation (900 ns) for an unrestrained simulation where lipids remained stably bound (*center-left*), an unrestrained simulation where lipids dissociated (*center-right*), and a restrained simulation (*right*). LRRC8 subunits, white; lipids, salmon; water surface, blue. To the left of each 900-ns snapshot are pore profiles determined across simulation frames. Pore radii (Å) are plotted as a function of z-position within the pore (Å), with pore lipids included (solid lines colored from blue to green with increasing simulation time) or excluded (gray lines) from the pore radius calculation. Pore profiles were calculated every 10 ns, as described above.

$20.9 \pm 3.9$) upon hypotonicity-induced cell swelling, consistent with wild-type LRRC8A:D currents (Fig. 5b, c and Supplementary Fig. 7). Introduction of T48D into LRRC8D alone was not sufficient to induce significant constitutive channel activity ($28.8 \pm 4.6$, n.s.). However, introduction of T48D into LRRC8A resulted in constitutively active channels with reduced fold activation upon swelling ($2.8 \pm 0.4$, $P = 0.0050$ compared to WT). Incorporation of the T48D mutation in both LRRC8A and LRRC8D subunits further reduced fold activation ($1.6 \pm 0.2$, $P = 0.0032$ compared to WT, $P = 0.0326$ compared to LRRC8A(T48D):D). The increased penetrance of the T48D mutation in LRRC8A compared to LRRC8D is consistent with LRRC8A:D VRACs containing more LRRC8A than LRRC8D subunits under these expression conditions. Altogether, from these data, we conclude that LRRC8A:D VRACs are constricted in the closed state by pore lipids.

In summary, we resolved two conformations of a LRRC8A:D VRAC, revealing a hexameric assembly with high structural variability in the cytoplasmic LRR domains and linker regions. These structures show how incorporation of LRRC8D subunits increases hydrophobicity and expands the selectivity filter, which, in conjunction with N-terminal sequence differences, may contribute to an increased selectivity of LRRC8D-containing VRACs for larger and non-anionic substrates. However, mutagenesis experiments, especially using uncharged substrates, are required to definitively test the contribution of these residues to any selectivity differences. On the other hand, the structural determinants of other channel properties, including single-channel conductance and open probability, could not be inferred from these structures. It is also unclear whether the two observed conformations represent different states along a gating trajectory, as we do not yet understand how different LRR arrangements can affect channel properties. Since both conformations are occluded by lipids, we interpret both structures to represent closed states. Whether and how these conformations interconvert during channel gating remains to be determined.

Lastly, we observed lipids bound within the channel pore and validated their role in channel closure using molecular dynamics simulations and electrophysiology recordings. What forces drive the removal of pore lipids and what paths lipids take to exit the channel during opening remain unclear. Resolving this will ultimately require an understanding of what signals or forces are sensed by the channel for activation and/or capturing an open state of the channel for structural studies, but our simulations identify one potential mechanism and route for lipid movement through the channel pore.

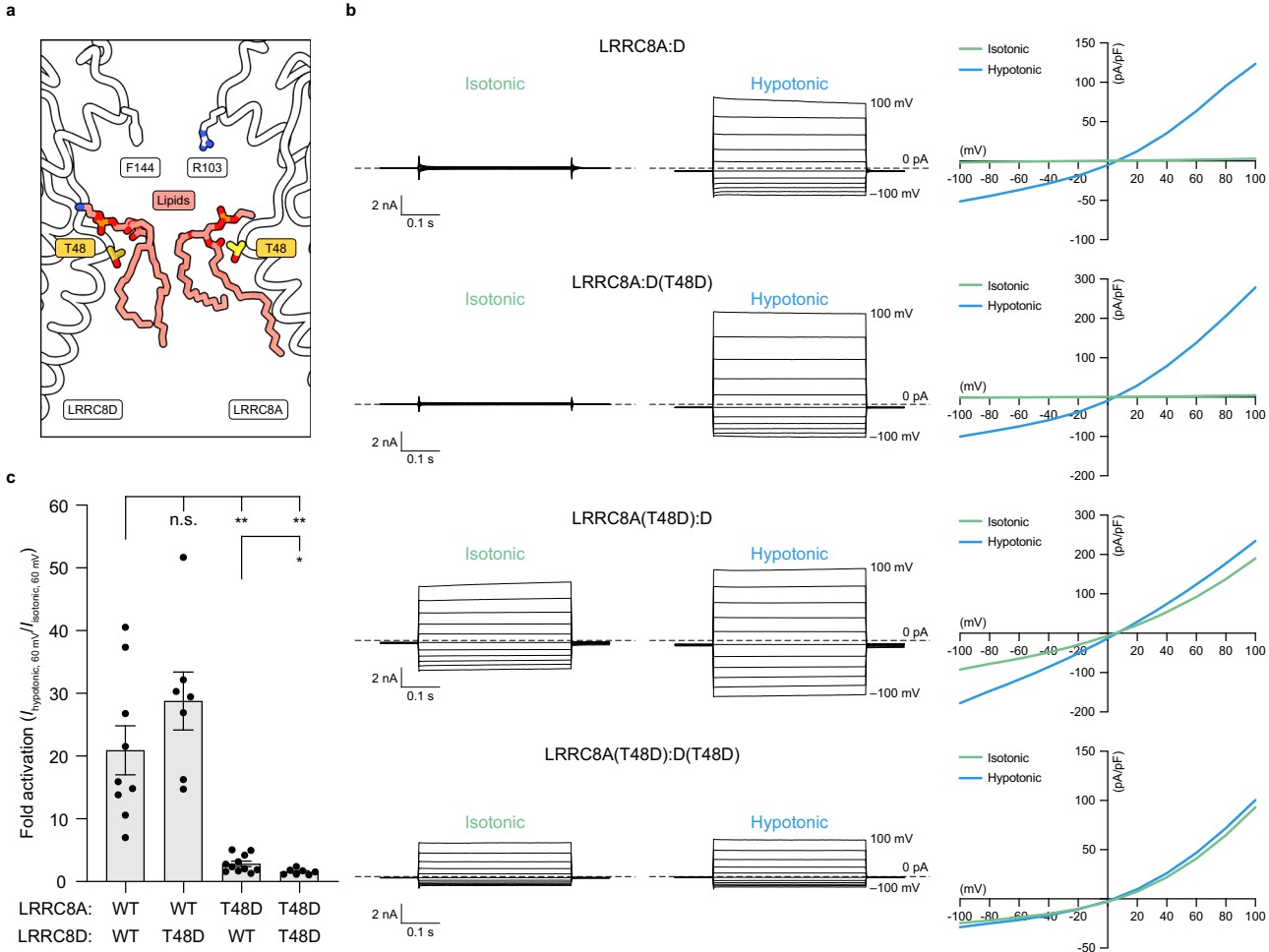

**Fig. 5 | LRRC8A:D VRACs are gated open upon removal of pore lipids. a** Focused side-view slice of the LRRC8A:D model pore with selectivity filter residues (white), pore lipids (salmon), and residue T48 (yellow) highlighted. **b** Whole-cell voltage-clamp recordings from *LRRC8A-E$^{-/-}$* HeLa cells expressing wild-type (WT) LRRC8A:D (*top*), LRRC8A:D(T48D) (*center-top*), LRRC8A(T48D):D (*center-bottom*), or LRRC8A(T48D):D(T48D) (*bottom*) mutant channels. For each construct, representative current traces from isotonic (*left*) and hypotonic (*center*) solutions are displayed alongside corresponding plots of the current-voltage relationships (*right*; isotonic, green; hypotonic, blue). For current traces, 0 pA/pF is marked with a dotted line. **c** Fold-activation of WT and mutant LRRC8A:D channels following hypotonic swelling ($I_{hypotonic,\,60\,mV}/I_{isotonic,\,60\,mV}$). Data are displayed as the mean ± s.e.m. plotted alongside individual data points for WT LRRC8A:D ($n = 9$), LRRC8A:D(T48D) ($n = 7$), LRRC8A(T48D):D ($n = 11$), and LRRC8A(T48D):D(T48D) ($n = 7$). Differences were assessed using a Brown-Forsythe and Welch one-way ANOVA test followed by a Dunnett's T3 multiple comparisons test to WT LRRC8A:D (n.s., not significant; **$P = 0.0050$ and $P = 0.0032$ for LRRC8A(T48D):D and LRRC8A(T48D):D(T48D), respectively) and a two-tailed unpaired $t$ test comparing LRRC8A(T48D):D to LRRC8A(T48D):D(T48D) (*$P = 0.0326$).

We speculate that wide intersubunit gaps capable of lipid permeation, as observed in LRRC8A:D conformation 1, may contribute to lipid gating by facilitating lipid entry and exit from the pore, especially if evacuation of structurally observed outer leaflet pore lipids and/or structurally plausible inner leaflet pore lipids is necessary for conduction. However, we were unable to observe lipid evacuation events in our simulations, which may be due to the short timescales of our simulations or an absence of the required forces or conformational rearrangements needed to drive lipid exit. Alternatively, our simulations showed that retaining lipids inside the pore in a displaced conformation is sufficient to open the conduction path, suggesting that full lipid evacuation may not be necessary for channel activity. Subunit N-termini are also expected to participate, but as they were not resolved in our structures, it is currently difficult to make predictions about their roles in lipid gating. Regardless, pore lipids have now been observed structurally and characterized electrophysiologically in LRRC8A:C heteromers[12], LRRC8A:D heteromers, and a LRRC8A-C chimera[25,27]. We thus expect pore lipids to be present in all VRAC assemblies and that lipid-gating is a general property of these channels.

## Methods

### Construct design and protein expression

The protein expression construct for LRRC8A$_{BRIL}$ was described previously[12]. Briefly, the *Mus musculus* LRRC8A sequence (UniProt: Q80WG5) was codon optimized for *Spodoptera frugiperda* and cloned into a custom vector based on the pACEBAC1 backbone (MultiBac; Geneva Biotech, Geneva, Switzerland) with an added C-terminal Pre-Scission protease cleavage site, linker sequence, superfolder GFP (sfGFP), and 7xHis tag. The sequence for apocytochrome b562RIL (BRIL)[14,28] was then codon optimized for *Spodoptera frugiperda* and inserted into the LRRC8A extracellular loop, between residues 76 and 91, generating a construct for expression of mmLRRC8A(76-BRIL-91)-SNS-LEVLFQGP-SRGGSGAAAGSGSGS-sfGFP-GSS-7xHis (LRRC8A$_{BRIL}$). The coding sequence for *Mus musculus* LRRC8D (UniProt: Q8BGR2) was codon optimized for *Spodoptera frugiperda*, synthesized (Gen9, Cambridge, MA), and cloned into a custom vector based on the pACEBAC1 backbone (MultiBac) with an added C-terminal TEV protease cleavage site, linker sequence, and mCherry tag, generating a construct for expression of mmLRRC8D-SNS-ENLYFQG-SRGSGSGS-mCherry. The LRRC8A and LRRC8D cassettes, each with a polyhedrin

promoter and SV40 terminator, were iteratively cloned using the I-CeuI and BstXI sites in the pACEBAC1 backbones to generate the dual LRRC8A_BRIL:D expression plasmid. MultiBac cells were then used to generate a LRRC8A_BRIL:D bacmid according to the manufacturer's instructions.

For electrophysiology, untagged versions of *Mus musculus* LRRC8A and LRRC8D were each cloned into mammalian expression vectors containing a cytomegalovirus (CMV) promoter and IRES eGFP or mCherry expression reporters, respectively. The LRRC8A(T48D) mutant construct was described previously[12]. The T48D mutation was introduced into LRRC8D by PCR using the following primers: GATG-CAACTGGATAAGGACCAGG (forward) and GTACCAGCGAATATAGC (reverse).

### Channel expression and purification
For protein production, Sf9 cells (Expression Systems, Davis, CA) were cultured in ESF 921 medium (Expression Systems) supplemented with 100 units/mL penicillin, 100 µg/mL streptomycin, and 0.25 µg/mL Amphotericin B (Gibco) and grown at 27 °C, shaking at 130 rpm. Baculovirus generation and amplification were conducted as described previously[12,29]. Briefly, P1 baculovirus was generated from cells transfected with LRRC8A_BRIL:D bacmid using Escort IV reagent (MilliporeSigma, Burlington, MA) according to the manufacturer's instructions. P2 virus was generated by infecting cells at 2 million cells/mL with P1 virus at a multiplicity of infection (MOI) of ~0.1. Infection was monitored by fluorescence, and cells were harvested at 72 h. P3 virus was generated in a similar manner to expand the viral stock. P3 virus was used to infect Sf9 cells at 4 million cells/mL at a MOI of ~2–5. At 72 h, infected cells expressing LRRC8A_BRIL:D were harvested by centrifugation at $2500 \times g$ for 10 min and frozen at −80 °C. Cell pellets for purification came from two batches of 1 L cultures (total: 2 L culture, ~28 mL cell pellet).

For protein purification (summarized in Supplementary Fig. 1), cell pellets were thawed and resuspended in 100 mL of lysis buffer (50 mM HEPES, 150 mM KCl, 1 mM EDTA, pH 7.4). Protease inhibitors (final concentrations: E64 (1 µM), pepstatin A (1 µg/mL), soy trypsin inhibitor (10 µg/mL), benzamidine (1 mM), aprotinin (1 µg/mL), leupeptin (1 µg/mL), AEBSF (1 mM), and PMSF (1 mM)) were added to the lysis buffer immediately before use. Benzonase (4 µl) was added after cell thaw. Cells were lysed by sonication and centrifuged at 150,000 x g for 1 h at 4 °C. The supernatant was discarded, and residual nucleic acid was removed from the top of the membrane pellet using D-PBS. Membrane pellets were scooped into a Dounce homogenizer and homogenized in 150 mL of extraction buffer (50 mM HEPES, 150 mM KCl, 1 mM EDTA, 1% (m/v) glyco-diosgenin (GDN: Anatrace, Maumee, OH), all protease inhibitors present in lysis buffer except PMSF, pH 7.4). The resulting membrane homogenate was gently stirred for 2 h at 4 °C to extract membrane proteins, followed by centrifugation at $30,000 \times g$ for 45 min at 4 °C. The supernatant, containing solubilized membrane proteins, was bound to 5 mL of pre-washed sepharose resin coupled to α-mCherry nanobody with gentle stirring for 1 h at 4 °C. The resin was then collected in a column and washed with 10 mL of buffer 1 (20 mM HEPES, 150 mM KCl, 1 mM EDTA, 0.02% GDN, pH 7.4), 40 mL of buffer 2 (20 mM HEPES, 500 mM KCl, 1 mM EDTA, 0.02% GDN, pH 7.4), and 10 mL of buffer 1. The resin was then resuspended with ~6 mL of buffer 1 containing 1.5 mM dithiothreitol (DTT) and 1 mg of TEV protease and nutated gently in the capped column overnight at 4 °C. Cleaved LRRC8 complexes were eluted with an additional ~7 mL of buffer 1. Eluate was then applied to a column containing 5 mL of sepharose resin coupled to α-GFP nanobody for a total of five passes through the resin. The resin was then washed with 20 mL of buffer 1, 10 mL of buffer 2, and another 10 mL of buffer 1. The resin was then resuspended with 6 mL of buffer 1 containing 0.5 mg of PreScission protease and nutated gently in the capped column for 2 h at 4 °C. Cleaved LRRC8A_BRIL:D heteromeric complexes were then eluted with

an additional ~7 mL of buffer 1, spin concentrated to ~500 µl with a 100 kDa molecular weight cutoff Amicon Ultra spin concentrator (Millipore), and then loaded onto a Superose 6 Increase column (GE Healthcare, Chicago, IL) equilibrated in buffer 1 on an NGC system (Bio-Rad, Hercules, CA) running Chromlab 6.0. Peak fractions containing LRRC8A_BRIL:D complexes were pooled and spin concentrated to <500 µL. Purified α-BRIL Fab BAG2 and α-Fab Nb[12,15,16] were thawed, diluted with buffer 1, and added in molar excess to LRRC8A_BRIL:D complexes at a ratio of ~1:1.5:3 (LRRC8: BAG2: Nb) and gently nutated at 4 °C for 30 min. The mixture was spin concentrated, centrifuged at $21,000 \times g$ for 5 min at 4 °C, and the supernatant (~8 µL at 1.9 mg/mL) was used immediately for cryo-EM sample preparation.

### Cryo-EM sample preparation
To prepare cryo-EM grids, 2 µL of sample was applied to a freshly glow-discharged (PELCO easiGlow, settings: 0.39 mBar, 25 mA, 25 s glow, 10 s hold) Holey Carbon 300 mesh R 1.2/1.3 gold grid (Quantifoil, Großlöbichau, Germany). After a ~5 s manual wait time, the grid was blotted (Whatman #1 filter paper) for 3 s at blot force 1 and immediately plunge frozen in liquid nitrogen-cooled liquid ethane using a Vitrobot Mark IV (Thermo Fisher Scientific) operated at 4 °C and 100% humidity. Grids were clipped after freezing. For grid preparation, the operator was wearing a mask.

### Cryo-EM data collection
All datasets were collected on a Titan Krios G3i electron microscope (Thermo Fisher) operated at 300 kV and equipped with a Gatan Bio-Quantum Imaging Filter with a slit width of 20 eV. Dose-fractionated images (~50 electrons per Å$^2$ applied over 50 frames) were recorded on a K3 direct electron detector (Gatan) with a pixel size of 1.048 Å. 242 movies were collected around a central hole position using image shift with an 11 × 11 hole pattern and two positions were targeted per hole. The defocus target was varied from −0.6 to −1.6 µm using SerialEM[30].

### Cryo-EM data processing
9730 movie stacks were collected, motion-corrected using MotionCor2[31] in RELION3.1[32], and CTF-corrected using CTFFIND 4.1.14[33] (Supplementary Figs. 2 and 3 and Supplementary Table 1 provide additional processing details). Micrographs with a CTFFIND reported resolution estimate greater than 4 Å were discarded, leaving 8,695 micrographs for further processing. An initial particle set of 579 particles was generated by manual picking in RELION and used to train Topaz[34] in RELION, which in turn was used to pick a set of 20,614 particles from 500 micrographs. These particles were cleaned to 6531 particles using 2D classification and ab-initio jobs in cryoSPARC[35]. Particles were transferred back to RELION using UCSF pyem tools[36] and used to train Topaz to pick particles from all 8695 micrographs, which were then cleaned in cryoSPARC using 2D classification, ab initio, and heterogenous refinement. The process of all-micrograph Topaz particle picking followed by particle cleaning in cryoSPARC was repeated iteratively for three total times, after which the three particle stacks were merged and de-duplicated in RELION with a 100 Å minimum inter-particle distance to generate a stack of 220,293 particles. These particles were non-uniform refined in cryoSPARC4.2.1[37], then transferred to RELION for post-processing and Bayesian polishing[38]. The polished particle stack was then further cleaned using 2D classification and heterogeneous refinement jobs in cryoSPARC to generate a final "consensus" particle stack of 120,274 particles. The consensus particle stack was non-uniform refined, then iteratively CTF refined (local, then global), followed by a final non-uniform refinement to give a 3.12 Å consensus map, which displayed marked heterogeneity in some subunits' linker regions and LRR domains.

To resolve this heterogeneity, we used 3D variability analysis and display jobs combined with heterogeneous refinement in cryoSPARC[18]. First, using the consensus map, we generated a mask to exclude

micellar density. We next applied this mask and the consensus particle stack to a four-component 3D variability analysis job with a 5 Å filter resolution. Next, using the 3D variability display job on simple mode (20 frames, 5 Å filter resolution), we identified two 3D variability components with pronounced conformational changes in subunit linker regions and LRR domains. We used the end volumes (i.e., frames 0 and 19) of these two components produced from the 3D variability display output, alongside a 5 Å low pass-filtered consensus map, as volume references for heterogeneous refinement of the consensus particle stack. Each class was then non-uniform refined, resulting in two well-resolved LRRC8A$_{BRIL}$:D classes (class 0: 3.47 Å, 27,322 particles; class 1: 3.50 Å, 29,827 particles), one class with less resolved LRRs (class 2, 3.48 Å, 26,507 particles), and two classes of significantly lower resolution (class 3: 8.57 Å, 19,744 particles; class 4: 8.44 Å, 16,874 particles). The particles from classes 2–4 were pooled together and applied to a second round of heterogeneous refinement, with the non-uniformed volumes from classes 0, 1, and 2 as reference volumes. Particles which sorted into classes 0 and 1 in the second round of heterogenous refinement were combined with the initial class 0 and 1 particle stacks and non-uniformed, giving rise to maps for two conformations of LRRC8A$_{BRIL}$:D: conformation 1 (3.29 Å, 43,907 particles, from class 0 particle stacks) and conformation 2 (3.35 Å, 46,893 particles, from class 1 particle stacks). A particle box size of 416 pixels was used throughout, and C1 symmetry was applied for all 3D jobs.

## Structure modeling, refinement, and analysis

LRRC8A and LRRC8D subunit identity was inferred from the positions of globular densities corresponding to inserted BRIL domains and confirmed with high-resolution features during modeling (Supplementary Fig. 4). Sharpened and unsharpened maps from cryoSPARC non-uniform refinement were used to build models in Coot[39]. As a starting point for LRRC8A:D conformation 1, one LRRC8A subunit from a structural model of LRRC8A$_{BRIL}$:C (PDB: 8DS3, chain B)[12] was trimmed (removing residues 412–808) and docked into the LRRC8A:D map for each of the six subunits. For the two LRRC8D subunits, the sequence was then mutated. For LRRC8A:D conformation 2, the LRRC8A:D conformation 1 model was used as a starting point. Models were real-space refined in Phenix[40] using sharpened maps and assessed for proper stereochemistry and geometry using MolProbity[41] (Supplementary Table 1).

The final LRRC8A$_{BRIL}$:D models consist of 1873 and 1880 amino acid residues for conformations 1 and 2, respectively. Unmodeled regions consist of each subunit's N-terminus (residues 1–14), extracellular loop 1 (residues 61–92, 69–92, or 69–93 for LRRC8A subunits, along with their inserted BRIL:α-BRIL Fab:α-Fab Nb complexes, residues 61–133 for LRRC8D subunits), intracellular loop (residues 175–231 for LRRC8A subunits, residues 217–276 for LRRC8D subunits), and LRR domain (residues 412–810 for LRRC8A subunits, residues 457–859 for LRRC8D subunits). In both conformations, 18 lipid molecules were modeled as DOPE (3/18 modeled completely), with ligand restraints obtained from the REFMAC monomer library using Coot.

For display purposes, we docked models for the best resolved LRRs in unsharpened maps using ChimeraX[42]. For LRRC8A subunits, we docked a model of a tightly interacting pair of LRRC8A LRRs obtained from a structural model of LRRC8A$_{BRIL}$:C (PDB: 8DS3, Chains B and C, residues 412–808)[12]. For LRRC8D subunits, we used the AlphaFold 3 Server[43] to predict a model of a LRRC8D monomer, which was trimmed to include residues 457–848 and docked. Two LRRC8A and two LRRC8D LRRs were docked for conformation 1; four LRRC8A LRRs were docked for conformation 2.

Measurements of pore constriction at the selectivity filter (Fig. 2) and pore lipid gate (Fig. 3) were made with HOLE[44] using protonated models. Figures were prepared using ChimeraX, Prism, and Adobe Illustrator software.

## Molecular dynamics simulations

We initiated simulations from the atomic model for LRRC8A:D conformation 1 (PDB: 9DX7). We retained the three pore-bound lipids and modeled them as DOPE; we removed the remaining lipid molecules from the peripheries of the protein structure. Prime (Schrödinger) was used to model hydrogen atoms and to add neutral acetyl and methylamide groups to cap protein termini. We retained titratable residues in their dominant protonation state at pH 7.4, resulting in protonation of D110 in chain D. To hydrate the vestibule immediately beneath the constriction point, we carried out an initial molecular dynamics simulation (using the protocols described below but with a shorter, 20-ns equilibration) to allow waters to flood into the vestibule while the protein remained restrained. We then used post-equilibration snapshots to extract these waters and add them back into the initial structure. We also used Dowser[45] to hydrate pockets near protein chains.

We used two different software protocols to place the prepared LRRC8A:D structure into a lipid bilayer and to create a simulation box. We reasoned that these different approaches could result in an increased diversity of initial configurations of bulk lipid interactions with the protein.

In the first strategy, we used PACKMOL-Memgen[46] within AmberTools (2025)[47] to generate a simulation box. Bilayers were composed of POPC in a 9:1 ratio with cholesterol. Sodium and chloride ions were added to neutralize the system to a concentration of 150 mM. Box dimensions were selected to maintain a 20 Å buffer between the protein image and the edge of the box, resulting in a simulation box of 145 Å × 146 Å × 164 Å and 391,424 atoms, including 'dummy' atoms employed in the four-point OPC water model. We used tLeap[47] to apply force-field parameters. We employed the ff19SB protein force field[48], the Lipid21 lipid force field[49], the four-point OPC water model, and corresponding 12-6-4 ion force field[50].

In the second strategy, we employed the CHARMM-GUI Membrane Builder tool[51,52] to prepare the simulation box and apply the same force-field parameters as those listed above. We also used CHARMM-GUI to protonate the cryo-EM structure with identical protonation of D110 in chain D and disulfide bond specification as used to prepare the system with Prime. Bilayers were composed only of POPC lipids and lacked cholesterol. Sodium and chloride ions were added to neutralize the system to a concentration of 150 mM. The simulation box was 140 Å × 140 Å × 166 Å and composed of 377,384–379,368 atoms.

For both initial configurations, we initiated simulations using the Compute Unified Device Architecture version of Particle Mesh Ewald MD using the AMBER24 software on single graphics processing units (GPUs)[53,54]. Systems were minimized in two stages, each composed of steepest descent minimization followed by conjugate gradient minimization. Systems were then heated from 0 K to 100 K in the NVT ensemble over 12.5 ps and then from 100 K to 310 K in the NPT ensemble over 125 ps at 1 bar, with harmonic restraints of 5.0 kcal mol$^{-1}$ Å$^{-2}$ placed on all atoms (including lipids), except for waters and ions. Systems were equilibrated at 310 K in the NPT ensemble at 1 bar using either of two restraint protocols over 60 ns: in the first protocol, restraints on protein and lipid atoms were tapered from 5.0 kcal mol$^{-1}$ Å$^{-2}$ by 1.0 kcal mol$^{-1}$ Å$^{-2}$ over 20 ns and then from 0.5 kcal mol$^{-1}$ Å$^{-2}$ by 0.1 kcal mol$^{-1}$ Å$^{-2}$ over 20 additional ns. An additional 20 ns of restrained equilibration were performed with only protein and pore lipids restrained with a force of 0.1 kcal mol$^{-1}$ Å$^{-2}$. In the second protocol, restraints on protein and lipid atoms were tapered from 5.0 kcal mol$^{-1}$ Å$^{-2}$ by 1.0 kcal mol$^{-1}$ Å$^{-2}$ over 25 ns; held at 0.5 kcal mol$^{-1}$ Å$^{-2}$ for 5 additional ns; then tapered again from 0.5 kcal mol$^{-1}$ Å$^{-2}$ by 0.1 kcal mol$^{-1}$ Å$^{-2}$ with protein and pore lipids restrained over 25 ns; and then held once more at 0.05 kcal mol$^{-1}$ Å$^{-2}$ with protein and pore lipids restrained for 5 additional ns. Production simulations were carried out in the NPT ensemble at 310 K and 1 bar,

using a Langevin thermostat for temperature coupling and a Berendsen barostat with semi-isotropic control for pressure coupling. For production simulations performed with restraints on the protein backbone, during equilibration, backbone restraints were held to 0.5 kcal mol$^{-1}$ Å$^{-2}$ while restraints on protein side chains and pore lipids continued to decrease per the protocols described above. Nonbonded interactions were cut off at 10.0 Å; long-range electrostatic interactions were calculated using Particle Mesh Ewald with an Ewald coefficient of 0.27511 and a B-spline interpolation order of 4. The FFT grid size was chosen such that the width of each grid cell was ~1 Å. We applied hydrogen mass repartitioning to use a 4-fs time step and constrained bond lengths to hydrogen atoms using SHAKE[55,56]. Trajectory snapshots were saved every 200 ps.

Simulation analysis was performed using Visual Molecular Dynamics (VMD)[57] and the MDAnalysis python package[58]. We carried out three separate analyses. (1) To quantify the number of lipids present in the pore, we considered a lipid bound if the lipid tilt angle with respect to the z-axis was >100° and the headgroup center of geometry z coordinate was above the T48 Cα atoms. (2) To measure constriction within the pore, we used CAVER 3.01[59] to identify permeation pathways from select simulation snapshots, with parameters of probe radius = 0.05, shell radius = 10.0, shell depth = 4.0, and maximum number of clusters = 999. (3) To measure compaction of the LRRC8D–LRRC8A and LRRC8A–LRRC8A subunit interfaces, we calculated the centers of mass of the set of helices composing the transmembrane regions of chains F and A (LRRC8D–LRRC8A distance) and of chains B and C (LRRC8A–LRRC8A distance). All production simulation data, downsampled every 1 ns, were included in our analysis of the number of bound lipids in the pore and water occupancy (prior to segmentation by lipid-bound state). Production data was downsampled every 10 ns for pore radii analysis with CAVER. For a small number of simulation frames, CAVER failed to identify contiguous pores, and we excluded these data from our traces and pore profiles. A table summarizing molecular dynamics simulation details is provided as Supplementary Table 2, and a molecular dynamics simulation checklist is provided as Supplementary Table 3.

## Electrophysiology

HeLa *LRRC8·E$^{-/-}$* cells[22] were cultured in DMEM (Gibco) supplemented with 10% FBS (Gibco), 100 units/mL penicillin, and 100 μg/mL streptomycin (Gibco) and grown at 37 °C in the presence of 5% CO$_2$. Trypsinized cells were deposited on 5 mm glass coverslips in a 6-well dish 6 h to 2 days prior to transfection. At least 30 min prior to transfection, wells were replaced with antibiotic-free growth media. Consistent with a prior report[60], we were unable to reproducibly obtain large wild-type VRAC currents with a 1:1 transfection ratio of LRRC8A and LRRC8D plasmids. Thus, for all transfections, a 1:5 mass ratio of LRRC8A and LRRC8D plasmids was used and mixed with FuGENE 6 at a 1:3 volume ratio diluted in serum- and antibiotic-free DMEM (typically, 0.5 μg LRRC8A and 2.5 μg LRRC8D mixed with 9 μL FuGENE 6 in 100 μL DMEM). Media was replaced 12–24 h post-transfection with growth media containing antibiotics. Patch clamp experiments were conducted 36–48 h post-transfection.

For patch clamp recording, coverslips were placed in a perfusion chamber at room temperature in isotonic bath solution (90 mM NaCl, 2 mM KCl, 1 mM CaCl$_2$, 1 mM MgCl$_2$, 10 mM HEPES, 10 mM glucose, adjusted to pH 7.4 with NaOH). Mannitol was used to adjust the solution osmolarity to ~330 mOsm (the approximate osmolarity of the cell culture media) as measured by a vapor pressure osmometer (VAPRO, Model 5600, ELITechGroup) or a freezing-point depression osmometer (Model 3320, Advanced Instruments). Borosilicate glass pipettes were pulled to a resistance of ~1.5–4 MΩ and filled with pipet solution (133 mM CsCl, 5 mM EGTA, 2 mM CaCl$_2$, 1 mM MgCl$_2$, 10 mM HEPES, ~4 mM MgATP, adjusted to pH 7.4 with CsOH and ~330 mOsm with mannitol). An Axopatch 200B amplifier connected to a Digidata 1550B digitizer (Molecular Devices) was used for data acquisition with

pClamp10.7 software. Analog signals were filtered at 1 kHz and sampled at 10 kHz. Pressure application from the patch pipette was accomplished with a high-speed pressure clamp (HSPC, ALA Scientific). Once a patch was achieved in the whole-cell mode and a stable whole-cell capacitance was measured, voltage families were recorded to monitor pre-swelling currents with the following voltage protocol applied every 46 s: $V_{hold}$ = 0 mV; $V_{test}$ = −100 to +100 mV, Δ20 mV, 400 ms. When initial currents stabilized, hypotonic bath solution (same components as isotonic, but adjusted to ~250 mOsm with mannitol) was exchanged into the chamber, and currents were monitored over the course of cell swelling until currents stopped increasing for multiple records or the patch broke. If a patch exhibited signs of leaking or sealing, a membrane test protocol was examined to either reseal, re-break-in, or discard the patch. Care was taken to make sure that the measured pressure was slightly negative (i.e., −1 mm Hg) after the whole-cell mode was achieved. Cells were monitored using a brightfield microscope and were selected for patching based on cell morphology (aiming for healthy interphase cells) and the presence of fluorescent reporters (GFP and mCherry). Recordings were not corrected for liquid junction potentials (measured as <3 mV).

Data was analyzed using Clampfit 10.7, Excel, and Graphpad Prism 10 software. To quantify current densities, the mean current density from the last 50 ms of each trace prior to the onset of capacitive currents was measured using Clampfit and normalized by the cell's measured capacitance. The displayed data are current densities at 60 mV (where VRAC current inactivation is minimal) averaged from three consecutive records after achieving stable currents in isotonic or hypotonic solution, and the ratio of these current densities was used to measure fold activation. For each cell, the reversal potential was measured in Prism as the x-intercept (voltage) of a linear fit of the current amplitudes at 0 mV and −20 mV from three averaged consecutive hypotonic records. For plotting representative current traces, a single isotonic and hypotonic record was chosen, the data were decimated 10-fold (to 1 ms sampling interval), and plotted in Prism. Prism was used for statistical analyses of fold activation. A Brown-Forsythe and Welch one-way ANOVA test followed by a Dunnett's T3 multiple comparisons test was used to compare all mutant channels to wild-type LRRC8A:D. An unpaired *t*-test was used to compare LRRC8A(T48D):D and LRRC8A(T48D):D(T48D).

## Reporting summary

Further information on research design is available in the Nature Portfolio Reporting Summary linked to this article.

## Data availability

Atomic coordinates are deposited in the Protein Data Bank (PDB) under accession codes 9DX7 (LRRC8A$_{BRIL}$:D conformation 1) and 9DXA (LRRC8A$_{BRIL}$:D conformation 2). Cryo-EM maps are deposited in the Electron Microscopy Data Bank (EMDB) under accession codes EMD-47282 (LRRC8A$_{BRIL}$:D conformation 1) and EMD-47283 (LRRC8A$_{BRIL}$:D conformation 2). Original micrograph movies and particle stacks are deposited in the Electron Microscopy Public Image Archive (EMPIAR) under accession code EMPIAR-12510. Simulation trajectories and viewing scripts are deposited in Zenodo [https://doi.org/10.5281/zenodo.16921648]. The source data for Figs. 3–5 and Supplementary Figs. 1, 3, 5–7 are provided as a Source Data file. Previously published models used in this manuscript were accessed from the PDB under accession codes 6G8Z (homomeric LRRC8A); 8DS3 (LRRC8A:C conformation 1); and 6M04 (homomeric LRRC8D). Source data are provided with this paper.

## Code availability

Analysis scripts to run CAVER 3 on simulation data can be found on Github [https://github.com/Latorraca-Lab/CAVER_PROTEIN_TUNNEL_PROCESSING].

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

## Acknowledgements

We thank Dan Toso, Ravindra Thakkar, and Paul Tobias for microscope and computational support at the Cal-Cryo facility. We thank the laboratory of Ardem Patapoutian for generously providing the *LRRC8* knock-out HeLa cell line. We thank the laboratory of Anthony Kossiakoff for providing the α-BRIL Fab BAG2 and α-Fab Nb. We thank members of the Brohawn laboratory for discussions and feedback on the manuscript. Simulations were performed on the C2B2 Research Computing Cluster made available through NIH award S10OD032433. This work was funded by The New York Stem Cell Foundation, a McKnight Foundation Scholar Award, and a Sloan Research Fellowship to S.G.B., NIGMS grants no. GM128263 to D.M.K. and GM148823 to N.R.L., and a Shurl and Kay Curci Ph.D. Scholarship and a Shurl and Kay Curci Ph.D. Fellowship to A.L.

## Author contributions

D.M.K. designed the fiducial tagging approach. D.M.K. and K.H. performed biochemical experiments and collected cryo-EM data. A.L. processed cryo-EM data, modeled and refined the structures, generated mutant constructs, performed electrophysiology experiments, and analyzed all presented data. C.A.S. and N.R.L. performed and analyzed molecular dynamics simulations. A.L., D.M.K., and S.G.B. conceived of the project. A.L. and S.G.B. wrote the manuscript with input from all authors.

## Competing interests

The authors declare no competing interests.
