## [Transparent Peer Review file · Nature Communications]

Assembly and lipid-gating of LRRC8A:D volume-regulated anion channels

Corresponding Author: Professor Stephen Brohawn

Version 0:

Reviewer comments:

Reviewer #1

(Remarks to the Author)

LRRC8/VRAC channels have important roles in cell volume regulation and in the transport of small organic molecules, including neurotransmitters. The study by Lurie et al. for the first time presents cryo-EM structures of mouse LRRC8A/D heteromeric channels. Several cryo-EM structures of LRRC8 channels have already been published previously. Initially, these studies focused on non-physiological homomeric channels, but more recently also structures of heteromeric VRAC channels were reported. Many aspects of VRACs remain poorly understood, including the important question of how a channel can be highly selective for Cl over Na, but also conduct organic substrates with different electric charges, an issue indirectly addressed in the present work.

The work is well-conducted, clearly presented and largely agrees both with the previously published structures of LRRC8A and D homomers, and with LRRC8A/C heteromeric channels. It provides limited new insights, but is a useful and necessary resource for future research and especially for isoform-specific drug discovery.

Specific comments:

While increased amplitudes of the T48D mutation are consistent with the concept of lipid gating, this model is difficult to unequivocally prove. It is possible that the lipids in the pore are an artifact of protein purification and simply coincide with the functional effects of a mutation in a pore-lining residue. MD simulations may strengthen this conclusion. Lipid gating model would suggest state-dependent accessibility of the pore for e.g. thiol reagents which may be small enough to enter the selectivity filter of the 'closed' state.

The present structure contains no resolved N-termini which are known to participate in the pore lining. While the present structure cannot allow new conclusions on the role of the N-terminus, the potential contribution of LRRC8D N-termini to substrate selectivity should be discussed. N-terminal sequences are not conserved among the paralogues, and LRRC8D differs from other LRRC8s in at least one position, previously implicated in affecting ion selectivity, namely Ala8 (10.1016/j.celrep.2023.112926). Such a discussion is also relevant in view of the role of LRRC8D in allowing the transport of small molecules.

Reviewer #2

(Remarks to the Author)

Heteromeric assembly allows VRACs to diversify their channel properties. While cryo-EM structures of LRRC8A:C heteromers have revealed important features, including lipid binding along the permeation pathway, the assembly mechanisms of other heteromeric channels remain unclear. In this study, Lurie et al. present the cryo-EM structures of LRRC8A:D heteromers in two distinct conformations. In each conformation, two LRRC8D protomers are aligned next to each other, destabilizing the LRR domains and widening the lateral fenestrations. The authors also model lipids within the pore, consistent with previous observations in LRRC8A:C VRACs. Based on the constitutive activity of the T48D mutant, which prevents lipids from entering the pore, they propose that LRRC8A:D VRACs are also gated by lipids.

Overall, the structures are of high quality and provide valuable insights into LRRC8A:D heteromers. However, the proposed

lipid-gating and ion selectivity mechanisms are largely based on visual inspection of the structures, making these potentially interesting ideas speculative. A few additional experiments could provide more compelling mechanistic insights, which are currently lacking in the field. Below are my specific comments:

1. Lack of functional validation for the proposed ion selectivity mechanism: LRRC8A:D VRACs exhibit distinct properties, including reduced Cl⁻ conductance and increased permeability to large, non-anionic substrates. While the structural data suggest that F144 plays a key role in these properties, no experimental validation is provided. If the proposed mechanism is correct, mutating the corresponding residue in other heteromers, such as LRRC8A:C, should make them behave like LRRC8A:D. Without such experiments, the proposed ion selectivity mechanism (Fig. 2) remains speculative.
2. Potential role of LRR domain destabilization in ion permeation: One of the notable features of LRRC8A:D VRACs is the destabilized LRR domains, which appear to influence the way protomers interact within the transmembrane (TM) region. This rearrangement could also affect permeant selectivity. To distinguish this effect from the proposed ion selection mechanism at the selectivity filter, it would be beneficial to compare ion selectivity and conductance among wild-type channels, LRRC8A homomers carrying the F144 mutation, LRRC8A:D heteromers with F144 mutations in all six subunits, and those with additional mutations at key residues such as R103 and L105.
3. Unclear mechanism of lipid gating: The lipid-gating mechanism is intriguing but remains speculative. While the pore-lining residues appear to accommodate lipid molecules, it is unclear how these lipids are displaced when the channel opens. What forces drive the removal of lipids from these binding sites? Further functional assays or molecular dynamics simulations could provide more insights into the gating mechanism.
4. Limited background on the T48D mutation: The T48D mutation is an important tool in this study, but insufficient background is provided. It would be helpful to clarify that the overall VRAC structure is not affected by this mutation and that the structure of the mutant channel lacks bound lipids inside the pore.
5. Unclear justification for lipid assignments in the pore: Although certain lipids are modeled within the pore, it is unclear how well these assignments fit the density map. Figure 3B does not provide sufficient detail to fully evaluate the placement of lipids. It would be beneficial to display each lipid density in a manner similar to Extended Data Figure 4.
6. Structural differences between Conformation 1 and Conformation 2: The two observed conformations appear to differ in their TM regions. Do these conformations represent different functional states of the channel? Additionally, what are the consequences of the differing stability in the LRR domains? Addressing these questions would enhance our understanding of how these structural changes relate to channel function.

Version 1:

Reviewer comments:

Reviewer #1

(Remarks to the Author)

Lurie et al addressed all of my concerns in their revised manuscript and I can recommend it for publication.

Reviewer #2

(Remarks to the Author)

The authors have adequately addressed the major concerns raised by the reviewers. The newly added MD simulation studies, in particular, provide much needed mechanistic insights into lipid-gating. This is an interesting paper, and I have no further comments.

Reviewer #3

(Remarks to the Author)

The manuscript by Lurie et al. presents the first cryo-EM structures of the heteromeric LRRC8A:D volume-regulated anion channel (VRAC), complemented by molecular dynamics (MD) simulations and electrophysiological characterization. The study provides important structural insights into the stoichiometry, assembly, and a conserved lipid-gating mechanism of this channel. The authors have adequately addressed several concerns raised in the initial review, particularly through the addition of MD simulations that bolster the lipid-gating hypothesis. The structural data are of high quality, and the manuscript is generally well-written. However, a few key issues regarding functional correlation and mechanistic depth remain and should be addressed before publication.

Major Comments:

1. Lipid-Gating Mechanism and MD Simulations. The addition of all-atom molecular dynamics simulations significantly strengthens the proposed lipid-gating model. The observation that pore lipids remain stably bound in backbone-restrained simulations but can dissociate in unrestrained ones is compelling. However, the mechanistic link between lipid evacuation and channel activation remains somewhat speculative. The authors propose that lateral gaps facilitate lipid exit, but no full

lipid exchange was observed in the simulations. It would be beneficial to more explicitly discuss the limitations of the current simulation timescales and the potential energetic or structural triggers (e.g., membrane tension, subunit rearrangement) required for complete lipid removal and channel opening. Speculating on how the observed conformational dynamics might couple to this process in the Discussion would add valuable insight.

2. Functional Correlation of Selectivity Filter. The structural analysis of the selectivity filter, particularly the role of LRR8D-F144 in increasing hydrophobicity and pore size, is well-described and logically presented. However, as noted by the authors in their response, direct functional validation of F144's role in substrate selectivity is lacking. The attempts to measure differences in glutamate and taurine permeability between LRR8A:C and LRR8A:D are honestly reported but inconclusive. The manuscript should therefore more explicitly state in the main text that the role of F144, while structurally plausible, remains a hypothesis requiring future functional validation, ideally with uncharged substrates and more robust assay systems. The added discussion on the potential contribution of N-terminal residues is appropriate, but the central claim regarding the selectivity filter would benefit from a clearer framing as a structural model awaiting functional confirmation.

3. Biological Significance of the Two Conformations. The two resolved conformations reveal interesting dynamics in the LRR and transmembrane domains. The authors rightly note that both appear to be closed states. The question of whether these represent distinct functional intermediates or simply conformational heterogeneity in the closed state remains open. While resolving this may be beyond the scope of the current study, the Discussion should briefly explore the potential functional implications of this observed flexibility. For instance, could one conformation be more primed for activation or lipid evacuation than the other? Linking this to the known requirement of non-LRR8A subunits for full channel activation could be a fruitful avenue for speculation.

Point-by-point response:

Reviewer #1 (Remarks to the Author):

LRRC8/VRAC channels have important roles in cell volume regulation and in the transport of small organic molecules, including neurotransmitters. The study by Lurie et al. for the first time presents cryo-EM structures of mouse LRRC8A/D heteromeric channels. Several cryo-EM structures of LRRC8 channels have already been published previously. Initially, these studies focused on non-physiological homomeric channels, but more recently also structures of heteromeric VRAC channels were reported. Many aspects of VRACs remain poorly understood, including the important question of how a channel can be highly selective for Cl over Na, but also conduct organic substrates with different electric charges, an issue indirectly addressed in the present work.

The work is well-conducted, clearly presented and largely agrees both with the previously published structures of LRRC8A and D homomers, and with LRRC8A/C heteromeric channels. It provides limited new insights, but is a useful and necessary resource for future research and especially for isoform-specific drug discovery.

Specific comments:

While increased amplitudes of the T48D mutation are consistent with the concept of lipid gating, this model is difficult to unequivocally prove. It is possible that the lipids in the pore are an artifact of protein purification and simply coincide with the functional effects of a mutation in a pore-lining residue. MD simulations may strengthen this conclusion. Lipid gating model would suggest state-dependent accessibility of the pore for e.g. thiol reagents which may be small enough to enter the selectivity filter of the 'closed' state.

We thank the reviewer for this suggestion and agree that MD simulations could provide important support for the model and strengthen the paper's conclusions. We carried out all-atom molecular dynamics simulations of LRRC8A:D under three conditions, each with six 1- μ s replicates. We note that our simulations create conditions that challenge pore lipid stability in two respects: (i) the low-resolution intracellular LRRs were not included in the simulated structures, an effect we expect to increase channel dynamics and promote channel opening, and (ii) the channel vestibule underneath the pore lipids was filled with water, though large intersubunit openings observed in our structures of LRRC8A:C and LRRC8A:D suggest that lipids could also partially occupy this space. The simulated structure also lacks N-termini, which could otherwise reduce the dielectric of the vestibule and further increase the favorability of lipids residing within the pore, as noted by the reviewer in the subsequent comment. We hypothesized that lipids retained as an artifact would have low affinity for the pore and consistently dissociate in simulation, whereas lipids that gate the pore should remain bound over hundreds of nanoseconds.

Two of the three simulation conditions, which had no backbone restraints applied but differed in their bulk lipid compositions, showed both behaviors: pore lipids dissociated away from their initial positions after tens to hundreds of nanoseconds in some replicates and remained bound in the pore for the duration of the 1- μ s simulation in other replicates. We wondered whether the observed dissociation events could occur because the simulated structure lacks the LRR domains, which may impose physical constraints on inter-subunit packing. In a third simulation condition performed with gentle restraints applied to protein backbone atoms, all three lipids remain bound within the pore for the duration of each replicate. Unrestrained simulations consistently show movement of channel subunits to seal the large LRRC8A-LRRC8D intersubunit gap, but these movements do not obviously disrupt pore lipid binding. Since we do not observe major differences between lipid-bound and lipid-dissociated conformations, we propose that other thermal motions in the protein in the absence of intracellular restraints can alter interactions between pore lipids, water and protein, leading to their stochastic dissociation. Altogether, these simulations show how lipids can reside within the pore on long timescales and support the model for lipid block.

We also agree that the lipid gating model suggests state-dependent accessibility and inhibition by small molecules. We have added a citation to a recent preprint (Yamada T., *et al. bioRxiv* (2025). <https://doi.org/10.1101/2025.02.24.639894>) that shows, as predicted by a lipid-gating model, that when applied extracellularly, a membrane-impermeable thiol-reactive agent, MTSET, inhibits a mutant LRRC8 channel incorporating an N-terminal cysteine residue only when applied during cell swelling when channels are open.

The present structure contains no resolved N-termini which are known to participate in the pore lining. While the present structure cannot allow new conclusions on the role of the N-terminus, the potential contribution of LRRC8D N-termini to substrate selectivity should be discussed. N-terminal sequences are not conserved among the paralogues, and LRRC8D differs from other LRRC8s in at least one position, previously implicated in affecting ion selectivity, namely Ala8 (10.1016/j.celrep.2023.112926). Such a discussion is also relevant in view of the role of LRRC8D in allowing the transport of small molecules.

We have added additional text and citations to emphasize this important point.

Reviewer #2 (Remarks to the Author):

Heteromeric assembly allows VRACs to diversify their channel properties. While cryo-EM structures of LRRC8A:C heteromers have revealed important features, including lipid binding along the permeation pathway, the assembly mechanisms of other heteromeric channels remain unclear. In this study, Lurie et al. present the cryo-EM structures of LRRC8A:D heteromers in two distinct conformations. In each conformation, two LRRC8D protomers are aligned next to each other, destabilizing the LRR domains and widening the lateral fenestrations. The authors also model lipids within the pore, consistent with previous observations in LRRC8A:C VRACs. Based on the constitutive activity of the T48D mutant, which prevents lipids from entering the pore, they propose that LRRC8A:D VRACs are also gated by lipids.

Overall, the structures are of high quality and provide valuable insights into LRRC8A:D heteromers. However, the proposed lipid-gating and ion selectivity mechanisms are largely based on visual inspection of the structures, making these potentially interesting ideas speculative. A few additional experiments could provide more compelling mechanistic insights, which are currently lacking in the field. Below are my specific comments:

1. Lack of functional validation for the proposed ion selectivity mechanism: LRRC8A:D VRACs exhibit distinct properties, including reduced Cl⁻ conductance and increased permeability to large, non-anionic substrates. While the structural data suggest that F144 plays a key role in these properties, no experimental validation is provided. If the proposed mechanism is correct, mutating the corresponding residue in other heteromers, such as LRRC8A:C, should make them behave like LRRC8A:D. Without such experiments, the proposed ion selectivity mechanism (Fig. 2) remains speculative.

We agree with the reviewer that functional validation is necessary to understand how and to what extent F144 contributes to creating LRRC8A:D VRACs' unique substrate selectivity profile. We attempted to establish an experimental approach for measuring differences in substrate selectivity of defined VRAC heteromers and their mutants; we detail these attempts below. However, we were unable to convincingly show substantial selectivity differences for two charged substrates (taurine and glutamate) between LRRC8A:C and LRRC8A:D VRACs, a prerequisite for evaluating changes from selectivity filter mutations. We believe further investigation is warranted, but requires other approaches and is better left to future dedicated studies. We have therefore adjusted the text throughout to reflect the limited evidence describing F144's role in channel selectivity, cite and discuss work on other determinants of selectivity, and highlight the importance of additional follow-up studies for functionally testing F144's role in LRRC8A:D substrate selectivity.

We measured substrate selectivity using patch clamp electrophysiology. While most substrates unique to transport by LRRC8A:D VRACs are electrically neutral, we selected two which we reasoned may show some difference in selectivity between LRRC8A:C and LRRC8A:D VRACs: glutamate and taurine. Glutamate was selected as it is a known VRAC substrate, it is of large size, and a previous study has shown a modest increased selectivity for aspartate by LRRC8A:D VRACs when compared to LRRC8A:C VRACs. On the other hand, taurine has been shown by multiple studies to be primarily transported by LRRC8D-containing VRACs and thus should show selectivity differences between LRRC8A:D and LRRC8A:C VRACs.

We transfected *LRRC8*^{-/-} HeLa cells to express LRRC8A:C and LRRC8A:D VRACs and recorded currents under bi-ionic conditions, replacing the majority of the anionic component in our conventional chloride-based recording solutions with glutamate or taurine on one side of the membrane. We tested both configurations for each anion in four separate conditions: i) chloride in and glutamate out, ii) glutamate in and chloride out, iii) taurine in and chloride out, and iv) chloride in and taurine out. To record currents carried by taurine, which is electroneutral at neutral pH, we adjusted taurine-containing solutions to pH 8.2.

No consistent differences in substrate selectivity were observed (Response Figure 1). Glutamate showed small differences between LRRRC8A:C and LRRRC8A:D that depended on the ionic configuration (i.e. when glutamate was applied extracellularly, LRRRC8A:D was more selective for glutamate than LRRRC8A:C, but the opposite was observed when glutamate was applied intracellularly). Taurine showed no significant selectivity differences between LRRRC8A:C and LRRRC8A:D with high cell-to-cell variability and ionic configuration-dependent results. Large shifts in the reversal potential over the course of recording a cell were frequently observed, an unanticipated confounding factor. It is unclear whether this is due to changes in channel selectivity, other channel currents, or effects from ion depletion and accumulation – an inherent problem with long bi-ionic recordings of channel currents (Li M., *et al. Nat Neurosci* (2015). <https://doi.org/10.1101/2025.02.24.639894>).

Response Figure 1. Summary of substrate selectivity measurements from whole-cell voltage-clamp recordings of *LRRRC8A-E^{-/-}* HeLa cells transfected with wild-type LRRRC8A:C (blue) and LRRRC8A:D (green) channels. **A)** Reversal potentials (mV) and **B)** permeability ratios (P_X/P_{Cl} where X = glutamate or taurine) are plotted as means with individual points shown. The major anions ("glut": glutamate, "taur": taurine) of the intracellular ("in") and extracellular ("out") solutions are indicated for each condition (i – v). Reversal potentials and permeability ratios are reported upon hypotonic swelling, except for the intracellular taurine configuration which resulted in currents in both isotonic (iv) and hypotonic conditions (v) due to the low ionic strength of the intracellular solution. For each condition, the solutions were as follows: i) intracellular: 140 mM CsCl₂, 2 mM CaCl₂, 1 mM MgCl₂, 5 mM EGTA, 10 mM HEPES, 4 mM MgATP, pH 7.4 (CsOH), ~330 mOsm (mannitol); extracellular: 90 mM monosodium glutamate, 2 mM potassium gluconate, 1 mM calcium gluconate, 1 mM magnesium gluconate, 10 mM HEPES, 10 mM glucose, pH 7.4 (NaOH), ~250 mOsm (hypotonic, mannitol) and ~330 mOsm (isotonic, mannitol), ii) intracellular: 140 mM monosodium glutamate, 2 mM CaCl₂, 1 mM MgCl₂, 5 mM EGTA, 10 mM HEPES, 4 mM MgATP, pH 7.4 (NaOH), ~330 mOsm (mannitol); extracellular: 90 mM NaCl, 2 mM KCl, 1 mM CaCl₂, 1 mM MgCl₂, 10 mM HEPES, 10 mM glucose, pH 7.4 (NaOH), ~250 mOsm (hypotonic, mannitol) and ~330 mOsm (isotonic, mannitol), iii) intracellular: same as intracellular for i; extracellular: 200 mM taurine, 2 mM KOH, 1 mM Ca(OH)₂, 10 mM HEPES, 10 mM glucose, pH 8.2 (NaOH), ~250 mOsm (hypotonic, mannitol) and ~330 mOsm (isotonic, mannitol), iv-v) intracellular: 240 mM taurine, 2 mM Ca(OH)₂, 5 mM EGTA, 10 mM HEPES, 4 mM MgATP, pH 8.2 (NaOH), ~330 mOsm (mannitol); extracellular: same as extracellular for ii. For i and iii, an agar salt bridge containing 1 M KCl connected the ground electrode to the extracellular solution. For iv and v, a micro-agar salt bridge containing 1 M KCl connected the patch electrode to the intracellular solution (Shao X.M. & Feldman J.L. Micro-agar salt bridge in patch-clamp electrode holder stabilizes electrode potentials. *J Neurosci Methods* **159**, 108–115 (2006)). Liquid junction potentials were calculated (i) or measured (ii-v) and used to correct reversal potentials. Permeability ratios were calculated using a modified Goldman-Hodgkin-Katz equation. For iii-v, the concentration of anionic taurine was estimated using the Hendersen-Hasselbalch equation, assuming a pKa of 8.74 for the taurine amino group (Cohn, E.J. & Edsall, J.T. *Proteins, amino acids and peptides as ions and dipolar ions*. (Reinhold Publishing Corporation, New York, 1943)).

Substrate selectivity differences between LRRRC8A:D and LRRRC8A:C/E may be most different for uncharged or zwitterionic organic substrates including cisplatin, myo-inositol, GABA, and taurine (in the neutral form, see: Schober A.L., *et al. J Physiol* (2017). <https://doi.org/10.1113/JP275053>). Due to these substrates' lack of overall charge, scintillation counting would be the preferred approach to assess their permeability. This experiment is challenging with our current setup because *LRRRC8^{-/-}* cells show low transfection efficiency

(often much lower than 10%), making effects from expression level and intrinsic channel properties difficult to distinguish. In future work, specific VRAC heteromers and mutants could be expressed by generating transgenic stable cell lines, but this falls outside the scope of this study.

2. Potential role of LRR domain destabilization in ion permeation: One of the notable features of LRRC8A:D VRACs is the destabilized LRR domains, which appear to influence the way protomers interact within the transmembrane (TM) region. This rearrangement could also affect permeant selectivity. To distinguish this effect from the proposed ion selection mechanism at the selectivity filter, it would be beneficial to compare ion selectivity and conductance among wild-type channels, LRRC8A homomers carrying the F144 mutation, LRRC8A:D heteromers with F144 mutations in all six subunits, and those with additional mutations at key residues such as R103 and L105.

We were unable to establish a reliable approach for characterizing substrate selectivity differences between LRRC8A:D VRACs and other heteromers and think further efforts are better suited for future studies. Please see our response to Comment 1 above.

3. Unclear mechanism of lipid gating: The lipid-gating mechanism is intriguing but remains speculative. While the pore-lining residues appear to accommodate lipid molecules, it is unclear how these lipids are displaced when the channel opens. What forces drive the removal of lipids from these binding sites? Further functional assays or molecular dynamics simulations could provide more insights into the gating mechanism.

We addressed this important issue with new molecular dynamics simulations, which provide several insights into lipid dissociation.

First, we observe dissociation only in simulations where the protein backbone is not restrained (i.e. can relax away from the starting structure). 10 of 12 unrestrained simulations show dissociation of one, two, or all three lipids from their starting position in the pore. Analyses of pore diameter during the simulations show that movement of two lipids is usually required to create a pore large enough for Cl⁻ conduction. In simulations where the protein backbone is gently restrained to the conformation seen in cryo-EM, lipids remain near their original positions and sterically seal the pore to permeable ions for the entire μ s, suggesting that rearrangements in protein conformation relative to the starting conformation lead to lipid destabilization and their stochastic dissociation. These motions may arise in our simulations due to the absence of the LRR domains and/or the absence of N-termini within the pore.

Second, during dissociation, hydrocarbon tails pack into hydrophobic grooves along the vestibule and slide down the helices while the polar headgroups flip down into the aqueous cavity (Response Fig. 2). We speculate that full dissociation may occur via lateral intersubunit gaps, since the tails of bulk lipids can also access these same gaps. Flipping of the headgroups may allow pore lipids to partition into the inner leaflet via this pathway. We raise these points in the text while noting the limitations of our simulation setup. We suggest additional experiments

in the future will be important to better understand forces and molecular mechanisms involved in delipidation and channel opening.

Response Figure 2. Representative simulation snapshots show lipid behavior in the vicinity of T48 (yellow). In simulations where lipids dissociate, head groups drop and, in some cases, flip, while lipid tails remain associated with hydrophobic grooves in the vestibule. The tails of bulk lipids (bright blue) and pore lipids (salmon) can access the same lateral openings, although we do not observe full lipid translocation in and out of the pore on the timescales of our simulations.

4. Limited background on the T48D mutation: The T48D mutation is an important tool in this study, but insufficient background is provided. It would be helpful to clarify that the overall VRAC structure is not affected by this mutation and that the structure of the mutant channel lacks bound lipids inside the pore.

We have edited the text to better explain our previous work with the T48D mutant and its effects on channel structure.

5. Unclear justification for lipid assignments in the pore: Although certain lipids are modeled within the pore, it is unclear how well these assignments fit the density map. Figure 3B does not

provide sufficient detail to fully evaluate the placement of lipids. It would be beneficial to display each lipid density in a manner similar to Extended Data Figure 4.

We have added images of the modeled pore lipids within map density for both conformations (Supplementary Figure 4D). In the text we also describe in more detail how lipids were modeled within the pore.

6. Structural differences between Conformation 1 and Conformation 2: The two observed conformations appear to differ in their TM regions. Do these conformations represent different functional states of the channel? Additionally, what are the consequences of the differing stability in the LRR domains? Addressing these questions would enhance our understanding of how these structural changes relate to channel function.

We agree that the different conformations observed in cryo-EM studies of LRRC8A:D and LRRC8A:C (Kern D.M. *et al.*, *Nat Struct Mol Biol* (2023). <https://doi.org/10.1038/s41594-023-00944-6>) channels, which include differences in TM regions, are intriguing. However, it is not obvious from the structures that they represent different functional states because all resolved structures are closed by pore lipids. For the same reason, we do not yet understand the consequences of differing LRR interactions and arrangements on channel function. We expect future work, including MD simulations initiated from different starting structures, could provide insight into potential functional differences between captured conformations. We have added additional text to detail this important point.

Please find below a point by point response to remaining reviewer comments.

Reviewer #3 (Remarks to the Author):

The manuscript by Lurie et al. presents the first cryo-EM structures of the heteromeric LRRC8A:D volume-regulated anion channel (VRAC), complemented by molecular dynamics (MD) simulations and electrophysiological characterization. The study provides important structural insights into the stoichiometry, assembly, and a conserved lipid-gating mechanism of this channel. The authors have adequately addressed several concerns raised in the initial review, particularly through the addition of MD simulations that bolster the lipid-gating hypothesis. The structural data are of high quality, and the manuscript is generally well-written. However, a few key issues regarding functional correlation and mechanistic depth remain and should be addressed before publication.

We thank the reviewer for their careful reading of the manuscript and have addressed the points raised with changes to the text.

Major Comments:

1. Lipid-Gating Mechanism and MD Simulations. The addition of all-atom molecular dynamics simulations significantly strengthens the proposed lipid-gating model. The observation that pore lipids remain stably bound in backbone-restrained simulations but can dissociate in unrestrained ones is compelling. However, the mechanistic link between lipid evacuation and channel activation remains somewhat speculative. The authors propose that lateral gaps facilitate lipid exit, but no full lipid exchange was observed in the simulations. It would be beneficial to more explicitly discuss the limitations of the current simulation timescales and the potential energetic or structural triggers (e.g., membrane tension, subunit rearrangement) required for complete lipid removal and channel opening. Speculating on how the observed conformational dynamics might couple to this process in the Discussion would add valuable insight.

We have more fully addressed this in the final paragraph. We note that additional forces that drive lipid removal remain unclear, that we did not observe full lipid evacuation, that longer simulations or additional forces may be required for full lipid exit from the pore, and that full lipid dissociation from the pore is not required for opening a pore large enough to conduct ions based on our simulations.

2. Functional Correlation of Selectivity Filter. The structural analysis of the selectivity filter, particularly the role of LRRC8D-F144 in increasing hydrophobicity and pore size, is well-described and logically presented. However, as noted by the authors in their response, direct functional validation of F144's role in substrate selectivity is lacking. The attempts to measure differences in glutamate and taurine permeability between LRRC8A:C and LRRC8A:D are honestly reported but inconclusive. The manuscript should therefore more explicitly state in the main text that the role of F144, while structurally plausible, remains a hypothesis requiring future functional validation, ideally with uncharged substrates and more robust assay systems. The added discussion on the potential contribution of N-terminal residues is appropriate, but the central claim regarding the selectivity filter would benefit from a clearer framing as a structural model awaiting functional confirmation.

We more explicitly state that mutagenesis experiments, especially using uncharged substrates, are required to definitively test the hypothesis that these differences in the selectivity filter contribute to differences in substrate selectivity between channel types.

3. Biological Significance of the Two Conformations. The two resolved conformations reveal interesting dynamics in the LRR and transmembrane domains. The authors rightly note that both appear to be closed states. The question of whether these represent distinct functional intermediates or simply conformational heterogeneity in the closed state remains open. While resolving this may be beyond the scope of the current study, the Discussion should briefly explore the potential functional implications of this observed flexibility. For instance, could one conformation be more primed for activation or lipid evacuation than the other? Linking this to the known requirement of non-LRRC8A subunits for full channel activation could be a fruitful avenue for speculation.

We have expanded discussion of the two conformations and state that whether they represent different states along a gating trajectory or interconvert during the gating cycle remain open questions to be addressed in future work.